# Stepwise oxygenation of the Paleozoic atmosphere

Alexander J. Krause [1], Benjamin J.W. Mills [1], Shuang Zhang [2], Noah J. Planavsky[2], Timothy M. Lenton[3] & Simon W. Poulton [1]

Oxygen is essential for animal life, and while geochemical proxies have been instrumental in determining the broad evolutionary history of oxygen on Earth, much of our insight into Phanerozoic oxygen comes from biogeochemical modelling. The GEOCARBSULF model utilizes carbon and sulphur isotope records to produce the most detailed history of Phanerozoic atmospheric $O_2$ currently available. However, its predictions for the Paleozoic disagree with geochemical proxies, and with non-isotope modelling. Here we show that GEOCARBSULF oversimplifies the geochemistry of sulphur isotope fractionation, returning unrealistic values for the $O_2$ sourced from pyrite burial when oxygen is low. We rebuild the model from first principles, utilizing an improved numerical scheme, the latest carbon isotope data, and we replace the sulphur cycle equations in line with forwards modelling approaches. Our new model, GEOCARBSULFOR, produces a revised, highly-detailed prediction for Phanerozoic $O_2$ that is consistent with available proxy data, and independently supports a Paleozoic Oxygenation Event, which likely contributed to the observed radiation of complex, diverse fauna at this time.

[1] School of Earth and Environment, University of Leeds, Leeds LS2 9JT, UK. [2] Department of Geology and Geophysics, Yale University, New Haven, CT 06520, USA. [3] Earth System Science Group, College of Life and Environmental Sciences, University of Exeter, Exeter EX4 4QE, UK. Correspondence and requests for materials should be addressed to A.J.K. (email: eeajrb@leeds.ac.uk)

Oxygen plays a vital role in planetary habitability, but the precise history of atmospheric oxygen on Earth remains a puzzle. The first rise in atmospheric $O_2$ concentration to appreciable levels around 2.45–2.32 Ga, termed the Great Oxidation Event (GOE), is well-defined due to the loss of mass-independent sulphur isotope fractionation as stratospheric ozone abundance increased[1]. The presence of fossilized charcoal (inertinite) in sediments younger than 419 Ma defines another oxygen threshold, indicating that $pO_2$ exceeded the levels required for combustion during this period[2–4]. However, the absence of fossil charcoal before 419 Ma does not necessarily point to lower atmospheric $O_2$, and may simply reflect the absence of readily-combustible fuel before the evolution of woody plants (lignophytes)[3].

A variety of geochemical methodologies have been developed and applied to constrain the redox state of the ancient oceans and atmosphere between the GOE and the Devonian period[5–15]. Several studies point to a Neoproterozoic Oxygenation Event[16], but geochemical proxies also show widespread anoxia in early Paleozoic oceans. However, while marine geochemical proxies provide insightful baseline observations, it is difficult to quantitatively infer atmospheric oxygen levels from reconstructions of ocean redox chemistry[17]. Nevertheless, such data may be utilized and enhanced by whole Earth system biogeochemical models.

Biogeochemical 'box models' calculate how atmospheric oxygen concentrations may have fluctuated over the Phanerozoic Eon (541-0 Ma), with a particular strength lying in the potential for high-resolution temporal reconstruction of variability in atmospheric oxygen levels. These models make predictions by estimating source and sink fluxes in the geological oxygen cycle, which are the burial, weathering and tectonic recycling of organic carbon and pyrite sulphur. This is possible because the residence time of $O_2$ in the surface system (i.e., atmosphere and ocean) is very long (>1 Myr). Box models for Phanerozoic oxygen can be computed in a number of ways with regard to the representation of elemental cycles, and how they calculate weathering and burial fluxes that transfer mass between the crust and surface system. The most prominent of these models, GEOCARBSULFvolc[18–21], hereafter referred to simply as GEOCARBSULF, provides a detailed $O_2$ reconstruction, as it uses the extensive carbon and sulphur isotope records from sedimentary rocks, as well as time-dependent, normalized Earth system parameters (such as river runoff) to derive changes to carbon and sulphur cycling[18]. Other prominent box models (e.g., COPSE[22], MAGic[23]) opt instead to calculate productivity and burial of organic C and pyrite through an internal scheme of nutrient delivery and recycling. This gives them greater power to interrogate possible drivers of Earth system change, but reduces the ability to make detailed predictions directly from known geological data. For example, the COPSE model's reconstructions of $O_2$ levels are in large part driven by the evolution of the terrestrial biosphere, through changes to the C:P ratio of biomass and rates of primary productivity on the continents, which is imposed in the model based on paleobotanical evidence[22,24,25]. The MAGic model[23] is similarly biosphere-driven, utilizing a dataset of the organic carbon content of the Phanerozoic sedimentary rock record[26] to derive a terrestrial organic carbon burial flux, rather than using isotopic data. Because these models are driven by broad, long-term changes, they produce $O_2$ curves that have less detail than GEO-CARBSULF. The GEOCARBSULF reconstructions, or other isotope-derived curves, are most commonly used beyond the immediate field of $O_2$ modelling, for example when considering $O_2$ impacts on animal evolution[27–30].

Despite differences in the way these models operate, GEO-CARBSULF, COPSE, and MAGic generally agree on the broad changes in atmospheric $O_2$ over the last ~400 Myrs, whereby $pO_2$

rose to a peak of around 25–30% atm in the Permo-Carboniferous (~300 Ma), and remained above ~18% atm thereafter[21–24,31,32]. This broad picture is also supported by the inertinite record[3]. In contrast, there is considerable uncertainty in atmospheric oxygen levels during the early Paleozoic. Here, GEOCARBSULF predicts roughly present levels of atmospheric oxygen, which are not borne out by other models[22,24]. More importantly, these predictions conflict both with proxy evidence for widespread anoxia, which indicate that $pO_2$ was below roughly half the present level[7,11,14], and with the growing body of evidence for a step-change in surface oxygen levels during the Paleozoic (Fig. 1). This problem is further highlighted by recent work updating carbon isotope inputs in GEOCARBSULF[30], which cause the model to predict even higher levels of oxygen in the early Paleozoic, with 37 and ~35% atm in the Cambrian and Silurian, respectively (see Fig. 1). These results suggest that the original GEOCARBSULF model produces increasingly unstable oxygen predictions during the Carboniferous period, back through to the Precambrian, as the $pO_2$ oscillations become larger.

To summarize, GEOCARBSULF is the most widely-used resource for reconstructing Phanerozoic oxygen concentrations, but the severe conflict with proxy records for the Paleozoic highlights that there may be a major problem in the model construction. Here we revisit GEOCARBSULF and challenge some of the underlying assumptions of the model, to ultimately produce a revised model and a new high-resolution reconstruction of Phanerozoic atmospheric $O_2$. Our new reconstruction differs considerably from the original and is compatible with available proxy records, supporting a step-change in $O_2$ during the Paleozoic.

## Results

**Paleozoic oxygen controls.** Box models calculate the concentration of atmospheric oxygen by estimating the geological source and sink terms. These sources and sinks are related to the cycling of carbon and sulphur: both cycles produce reduced species (photosynthetic organic carbon and pyrite sulphur derived from microbially-produced $H_2S$) which, when buried in sediments, leads to net oxygenation of the surface system, following Eqs. (1) and (2):[33]

$$CO_2 + H_2O \rightarrow CH_2O + O_2 \tag{1}$$

$$\begin{aligned} 2Fe_2O_3 + 16Ca^{2+} + 16HCO_3^- + 8SO_4^{2-} \\ \rightarrow 4FeS_2 + 16CaCO_3 + 8H_2O + 15O_2 \end{aligned} \tag{2}$$

When these sediments are oxidatively weathered—usually following atmospheric exposure due to uplift—or are subject to metamorphism/diagenesis, the reverse of Eqs. (1) and (2) takes place, and pyrite sulphur and organic carbon are oxidised, thus decreasing the amount of free oxygen in the system. Ultimately, the concentration of atmospheric oxygen is determined by the rates of burial of organic C and pyrite S, versus the consumption of oxygen by the weathering of these same reduced species over geologic time[34].

The GEOCARBSULF model[18,19], and its predecessors[33,35], calculate these burial rates using isotope mass balance (IMB). Both photosynthesis and microbial sulphate reduction bestow significant isotopic fractionation effects, allowing their rates to be 'inverted' from the known isotopic compositions of sedimentary carbonates and sulphates. However, a major uncertainty in these models is the degree to which these isotopic fractionation effects are dependent on global $O_2$ levels. Such a dependence of

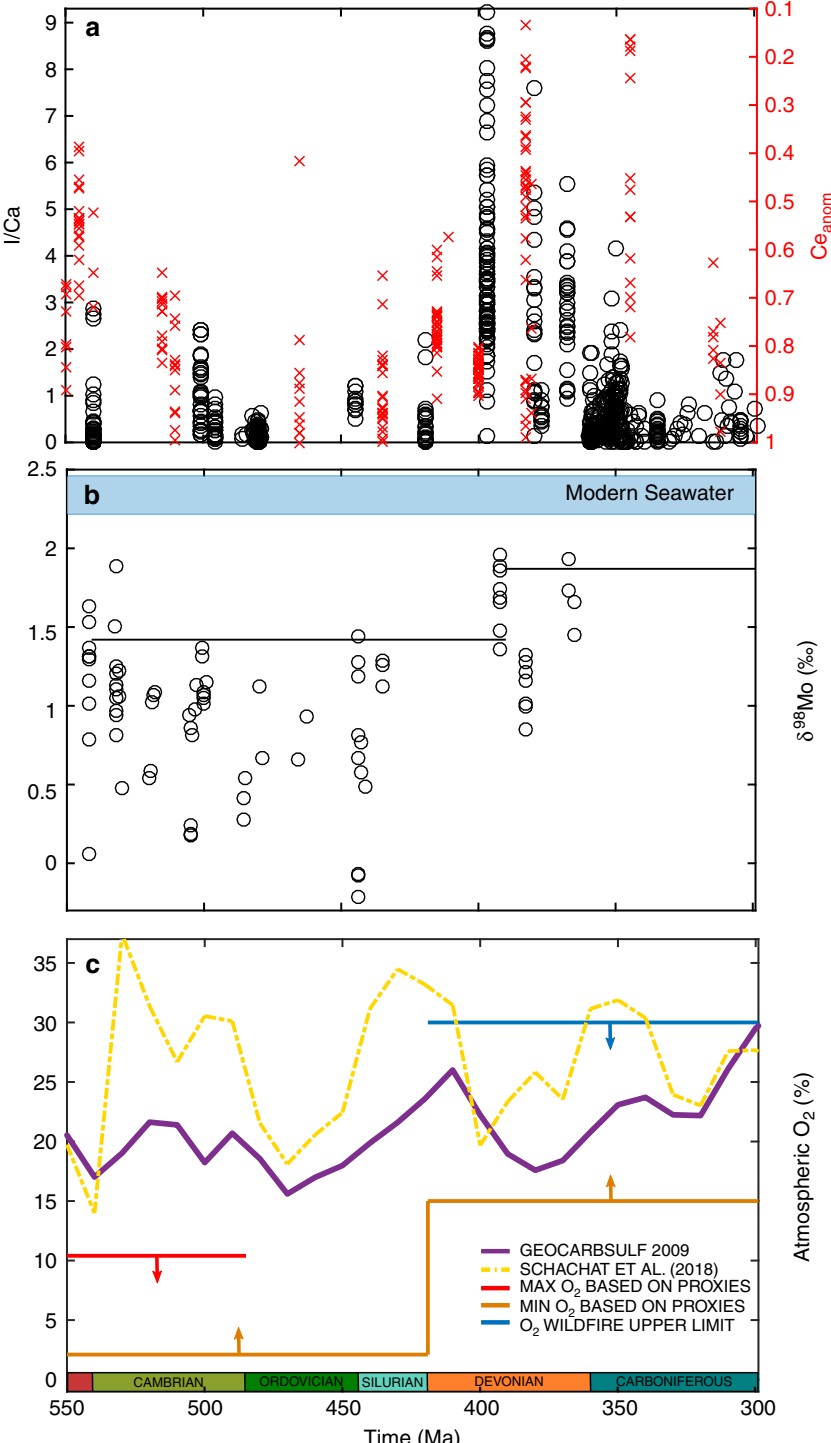

**Fig. 1** Atmospheric O$_2$ levels during the early-mid Paleozoic predicted by GEOCARBSULF, compared to geochemical proxy data. **a** Iodine to calcium ratios[50], and cerium anomaly data[15], taken from marine carbonates. Note: higher iodine to calcium ratios, and lower cerium anomaly values indicate oceanic oxygenation. **b** Molybdenum isotope values for the early to mid-Paleozoic, where the solid black lines represent 90% percentiles for the two time periods[11]. **c** The purple line represents the atmospheric O$_2$ output from GEOCARBSULF[21], the yellow dot-dash line shows recent work to update the δ$^{13}$C record in GEOCARBSULF[30], and the red line shows the approximate maximum atmospheric O$_2$ (10.4% atm) for the Cambrian and earlier, based on geochemical water column redox data and allowing for modelling uncertainties[14,72,73]. The orange line is the approximate O$_2$ minimum, based on the same redox data, as well as Cambrian biota and wildfire oxygen requirements[2,4,14]. The blue line is a likely oxygen maximum, based on wildfire feedbacks, but geochemical mass balance studies suggest $p$O$_2$ levels as high as 35% atm may be permissible[4,68,74]

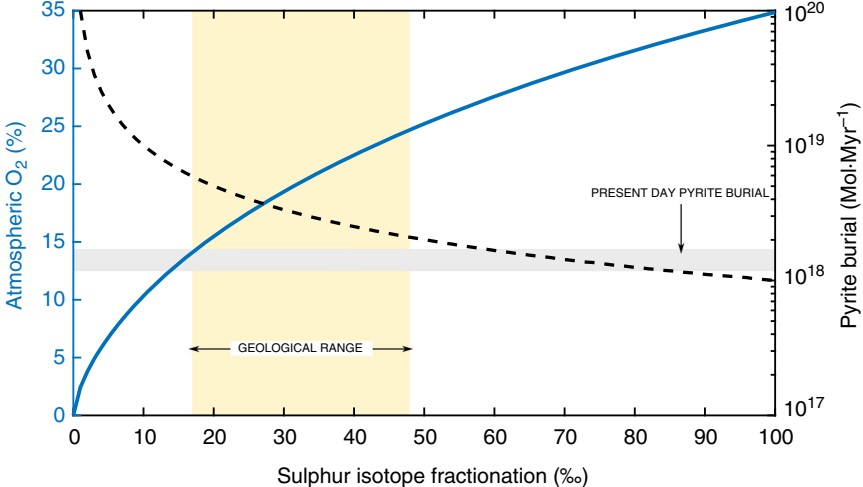

**Fig. 2** The relationship between sulphur isotope fractionation and oxygen concentration in GEOCARBSULF. The blue line shows the sulphur isotope fractionation ($\alpha_s$) generated by the equation used in the original GEOCARBSULF model for different concentrations of atmospheric $O_2$, and the dashed black line shows the resultant pyrite burial flux. The yellow box represents the bounds of the geologic record for sulphur isotope fractionation between sulphate and pyrite over the last 570 Myrs, based on datasets from Wu et al.[66] and the grey box represents estimated modern day pyrite burial levels[75,76]

fractionation on oxygen concentration provides stabilizing negative feedback in the model, whereby higher $O_2$ levels imply a higher degree of carbon and sulphur isotopic fractionation, which decreases the rate of organic C and pyrite burial calculated by the inversion approach, as less organic C or pyrite needs to be buried to achieve the same change in global isotope ratios.

The fractionation between carbonates and organic carbon is reasonably well understood from plant and plankton growth experiments and theory[36–38], and thus a simple equation fitted to the data is possible. By contrast, the fractionation of sulphur isotopes between sulphate and sulphide ($\alpha_s$) due to bacterial sulphate reduction, is a complex process. Only ~5–20% of sulphide produced by sulphate reduction in shelf sediments is permanently buried as pyrite, with the remainder subject to disproportionation or reoxidation to sulphate via numerous byzantine pathways, with each step likely impacting upon the isotopic signature[35,39–41].

For GEOCARBSULF, the sulphur isotope system was simplified and a sulphur isotope fractionation equation was derived (see supplementary equation (56)) based on the simple observation of larger fractionations at higher $O_2$ levels[42], with the underlying principle being that recycling of sulphur via reoxidation and disproportionation produced the high sulphur fractionations that were not observed in existing laboratory experiments of bacterial sulphate reduction[43–45]. More recent experimental studies (e.g., Sim et al.[46]) have, however, shown that large sulphur isotope fractionations can be obtained from microbial sulphate reduction alone, without the need for disproportionation, although the significance of this contribution to the global sulphur isotope fractionation record over geologic time remains unclear. Furthermore, as Jørgensen and Nelson[39] note, pyrite can be oxidized in anoxic settings through abiotic pathways (e.g., via manganese oxides), and thus the disproportionation pathway for large sulphur isotope fractionation is not necessarily dependent on higher $pO_2$ levels. Additionally, the source of sulphate and its isotopic signature are important to know, in order to derive fractionation values. While most sulphate in marine settings comes from bottom water sulphate, some fraction comes from the oxidation of organic sulphur compounds in particulate organic matter[47]. A lot of work has gone into elucidating the cryptic nature of the marine sulphur cycle, but it remains difficult to determine a quantitative relationship between oxygen

concentration and sulphur isotope fractionation on a global scale, due to the nature of this cycling, which may have also changed over geologic time. Consequently, the equation used in GEOCARBSULF contains a considerable amount of uncertainty.

Crucially, the formulation chosen results in a very strong $O_2$ feedback from the sulphur cycle in GEOCARBSULF. As $pO_2$ levels begin to decrease, the sulphur isotope fractionation value calculated by the model also decreases. This smaller fractionation results in an increase in pyrite burial calculated from IMB, and subsequently, a rise in $O_2$ production (see Fig. 2). Atmospheric oxygen levels <10% atm therefore result in an unrealistically low sulphur isotope fractionation (less than 10‰), and generate extremely large (~1–2 orders of magnitude higher than present day) amounts of pyrite burial. The concentration of atmospheric oxygen is thus tightly constrained by the sulphur cycle in the model, resulting in Paleozoic $pO_2$ values that are close to present day (see Fig. 1), with low values of atmospheric oxygen being unobtainable.

**Model development**. We update the GEOCARBSULF model to incorporate a process-based sulphur cycle that does not depend on inverting the $\delta^{34}S$ isotope record to calculate fluxes, and therefore does not rely on estimating oxygen-related changes in the fractionation factor. To do this we replace the equations which calculate the weathering and burial rates of both reduced (i.e., pyrite) and oxidized (i.e., gypsum) sulphur species with those used in the COPSE model[24]. In this modified system, the $O_2$ source from pyrite burial depends on the concentration of oceanic sulphate, the supply of organic matter to sediments, and the average oceanic dissolved oxygen concentration, thus retaining some level of $pO_2$-dependent feedback (see 'Methods' and Supplementary Note 3 for formulations). We also make modifications to the model which follow previous work: the carbon isotope record used to drive the model is updated to the latest compilation[32,48], and oxidative weathering of fossil organic carbon is assumed to be sensitive to $pO_2$[49]. Rather than alter existing code, we build the new model from scratch in MATLAB, incorporating a variable-step, variable-order, stiff ODE solver which improves the resilience of the model. We name the new model GEOCARBSULFOR, indicating that it uses a 'forwards' sulphur system, wherein sulphur cycle dynamics are calculated from other

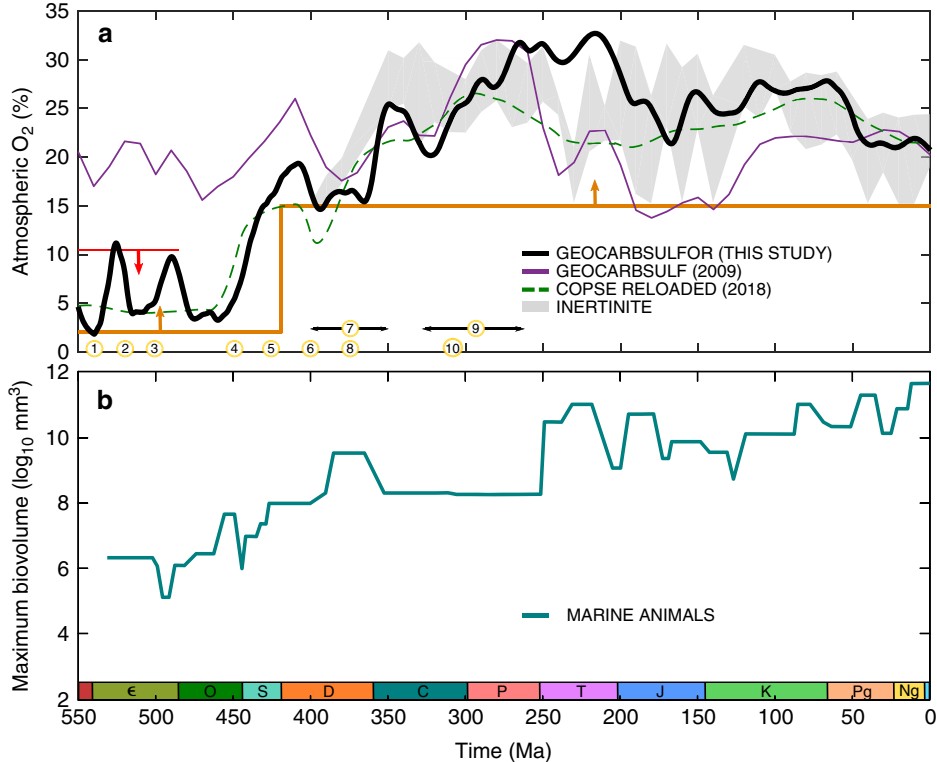

**Fig. 3** Results of GEOCARBSULFOR compared to GEOCARBSULF, the COPSE model, and to geochemical proxies. **a** The black line represents the results from GEOCARBSULFOR, the purple line is the original GEOCARBSULF[21] and the green dashed line is COPSE[22]. The grey envelope shows $pO_2$ predictions from inertinite data[3], and the arrowed lines show the boundaries from Fig. 1. The numbered circles denote: (1) the beginning of the Cambrian explosion and a decrease in cerium anomaly values, (2) bioturbation intensification, (3) the Steptoean positive carbon isotope excursion event, (4) the Great Ordovician Biodiversity Event and evidence of the first land plant spores, (5) earliest evidence for vascular land plants, (6) initial evidence for a rise in $\delta^{98}Mo$ and I/Ca and a decrease in cerium anomaly values, (7) a reduction in abundance of fossilized charcoal, (8) the first forests, (9) widespread mires and (10) a collapse in coal forests, and a rise to dominance of tree ferns[4,15,50,51,53,55,56,61]. **b** Marine animal body size[54] evolution over the Phanerozoic. Є Cambrian, O Ordovician, S Silurian, D Devonian, C Carboniferous, P Permian, T Triassic, J Jurassic, K Cretaceous, Pg Paleogene, Ng Neogene

model processes, instead of the $\delta^{34}S$ record. The isotope record is used instead for model validation. See 'Methods' for full details and model equations for the amended sulphur system, and Supplementary Notes 2 and 3 for the full list of model parameters and equations.

**Model predictions.** Our GEOCARBSULFOR model generates significantly different predictions for Phanerozoic $pO_2$ levels, relative to those generated by the original GEOCARBSULF model (Fig. 3). This is particularly evident in the early Paleozoic, where $O_2$ levels from GEOCARBSULFOR are much lower than the modern-day levels predicted by the original, and generally agree with geochemical redox proxies for this time period[11,14,15,50]. A number of $O_2$ variations in the new model can be linked to geobiological events, denoted with numbers in Fig. 3. The model calculates a transient rise in oxygen coincident with the Cambrian explosion (#1), which relaxes as bioturbation intensifies during Cambrian Stages 2 to 4[51] (#2). Bioturbation likely resulted in the increased burial of bioavailable phosphorus in sediments, thus limiting primary productivity and oxygen production, and exposed organic carbon and pyrite buried in anoxic sediments, to the overlying oxic water column, which led to oxidation[51,52]. A second transient increase in $pO_2$ begins around 500 Ma, driven in the model by the Steptoean positive carbon isotope excursion (#3), which suggests high rates of organic carbon burial and subsequently oxygen production[53].

GEOCARBSULFOR predicts a clear rise in $pO_2$ during the mid-Paleozoic at the time of the Great Ordovician Biodiversity

Event (#4). At roughly the same time, the maximum biovolume of marine animals began to increase from $10^6$ mm$^3$ to >$10^9$ mm$^3$ in the Devonian (Fig. 3b), which ties in with the radiation of large predatory fish that had increased oxygen demands[11,54]. While atmospheric $O_2$ concentration oscillated between ~2 and 11% atm during the Cambrian to mid-Ordovician, the mid-Paleozoic rise (the timing of which is in broad agreement with recent work[7,22,25,55],) to near modern-day levels in the Devonian coincides with the rise of vascular plants (#5). This marked a transition to an Earth system with $pO_2$ levels generally above 15% —the level required to sustain smouldering fires[2] and a level complemented by the results from several novel geochemical proxies (#6)[11,15,50]. A nadir of ~14.5% atm in the mid-Devonian (across the Emsian-Eifelian boundary at ~393 Ma) and generally lower $pO_2$ predictions during the mid-Emsian to end Famennian stages (~400–360 Ma) are compatible with the scarcity (but not total absence) of fossilized charcoal (#7) during this period, and with both modelling and experimental work exhibiting the low probability of achieving combustion when $pO_2 < 19\%$ atm[2,4,56,57].

GEOCARBSULFOR, and the original GEOCARBSULF, are driven primarily by the carbon isotope record, and as such, the model distinguishes a major change in the carbon cycle from the Ordovician to Carboniferous periods, which we attribute to the expansion of land plants and the rise of the first forests (#8), due in part, to terrestrial biomass having a much greater C:P ratio than marine organic matter[24]. This is consistent with the biosphere driven method of COPSE[22], which in a recent iteration of the model, assumes an increase in total organic carbon burial

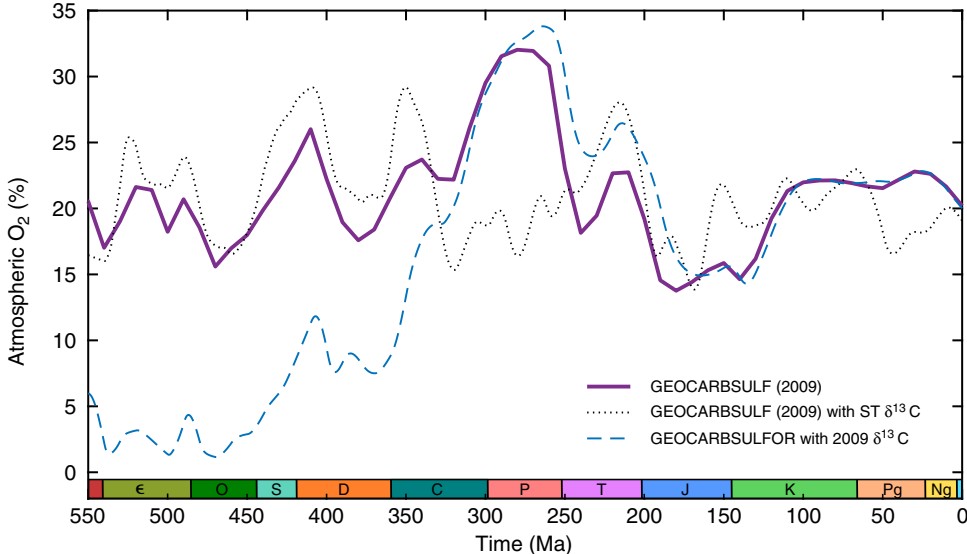

**Fig. 4** Sulphur cycle versus carbon isotope modifications. The purple line is the original GEOCARBSULF $p$O$_2$ prediction[21], the black dotted line is the original GEOCARBSULF model but with the δ$^{13}$C record updated to use data from Saltzman and Thomas[48], and the blue dashed line is GEOCARBSULFOR but with the isotope data from the original GEOCARBSULF model

with the rise of land plants from ~470 Ma onwards, generating a rise in $p$O$_2$ at this time (green dashed line in Fig. 3). The MAGic model, on the other hand, assumes that land plant material makes a relatively small contribution to total organic carbon burial today —which contrasts with data suggesting a large contribution[58,59]. Therefore, in MAGic, land plant evolution has little effect on organic carbon burial, and thus does not replicate this oxygenation event[23].

In the mid-Carboniferous (Bashkirian, 323 Ma), the $p$O$_2$ output from GEOCARBSULFOR briefly dips to ~20%, having been >25% in the early Mississippian. This decrease generally matches a drop in sea level at the time[60], which possibly exposed previously buried organic C to oxidative weathering. Following this, $p$O$_2$ levels rise once more to >20%, and this is concurrent with the appearance of widespread mires (#9), firstly in Euramerica, and then in Gondwana, Cathaysia and Angara, which provided ideal conditions for enhanced preservation of buried organic C[4,57]. Towards the end of the Carboniferous there was a collapse in the coal forests (#10), and opportunistic tree ferns took over as the dominant land plant species, probably due to aridification[61]. Despite this change to vegetation, $p$O$_2$ levels in our model do not drop, as swamp and mire conditions remained prevalent until near the end of the Permian, and thus organic C continued to be preserved[4].

The model results match well those from the oxygen proxy from inertinite data[3]—which back calculates $p$O$_2$ based on charcoal abundance in mire settings—for most of the Phanerozoic, and provides a better fit to this proxy than the original GEOCARBSULF. However, there is a large discrepancy between GEOCARBSULFOR and this inertinite proxy, and between both GEOCARBSULFOR and GEOCARBSULF/COPSE, in the Triassic. The original GEOCARBSULF predicts a rapid decline in $p$O$_2$ at the Permian-Triassic boundary, and while no early Triassic coals have yet been found, the inertinite proxy suggests that $p$O$_2$ would have declined substantially (to ~18.5%) by the mid-Triassic[3]. Yet, GEOCARBSULFOR suggests that atmospheric oxygen was around 30% for this entire period.

There are two reasons for this conflict: first, the isotope record used by the original GEOCARBSULF indicated a ~4‰ decrease in δ$^{13}$C at the end of the Permian, whereas the more recent records used for GEOCARBSULFOR show a decrease of only

~1.5‰. As such, there is a rapid decline in the amount of organic carbon buried in GEOCARBSULF, which is not matched to the same degree in GEOCARBSULFOR. Uncertainties surrounding the carbon isotope record are discussed further below. Secondly: pyrite burial in GEOCARBSULFOR is now partially dependent on the amount of sulphate in the ocean. During the Triassic, GEOCARBSULFOR predicts—and is validated by geochemical proxies and other modelling work (see Supplementary Fig. 4)— quite high levels of sulphate, with only a moderate decline over this period. This means pyrite burial rates for GEOCARBSULFOR do not decrease as substantially as they did in GEOCARBSULF, as sulphate levels buffer, to a degree, the changes to the organic carbon burial rates at this time. In the latest version of COPSE[22], the C:P land plant stoichiometry was updated, to factor in a decrease in C:P (from 2000:1 to 1000:1) over 345–300 Ma, and a new forcing: coal basin depositional area, was introduced, with a high depositional area in the Carboniferous and a sharp decline at the end of the Permian. These forcings in the COPSE model serve to keep predicted $p$O$_2$ levels much lower than those by GEOCARBSULFOR, over the end of the Permian and the Triassic.

## Discussion

The key differences between the Paleozoic predictions of GEOCARBSULFOR and the original GEOCARBSULF model arise from the modified sulphur cycle, rather than the new isotope data. We tested this by inputting the new δ$^{13}$C record, taken from Saltzman and Thomas[48], into the original GEOCARBSULF model, and then using the δ$^{13}$C record from GEOCARBSULF[21] to drive GEOCARBSULFOR (see Fig. 4). Using the new C isotope data in the original model results in early Paleozoic $p$O$_2$ levels which are broadly similar to those from the original model. However, using the older isotope record in our new model produces low $p$O$_2$ in the early Paleozoic, followed by near identical results from the mid-Carboniferous onwards. This leads us to conclude that atmospheric oxygen levels in the early Paleozoic are highly dependent on the sulphur cycle, although the carbon cycle helps to constrain some of the timings and magnitude of changes to $p$O$_2$.

As our model computes the sulphur cycle mechanistically, rather than relying on isotope inversion, it can output a synthetic

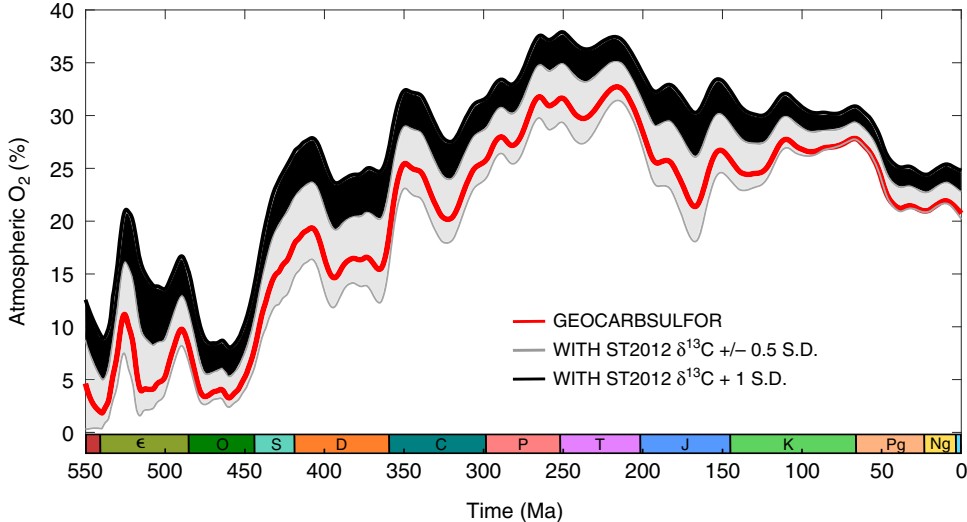

**Fig. 5** Atmospheric O$_2$ sensitivity to changes in the δ$^{13}$C record. The red line is our GEOCARBSULFOR model, the grey envelope is the atmospheric O$_2$ generated by ±half a standard deviation change to the ocean-atmosphere δ$^{13}$C record, based on the Saltzman and Thomas[48] data, and the black envelope is the +1 standard deviation

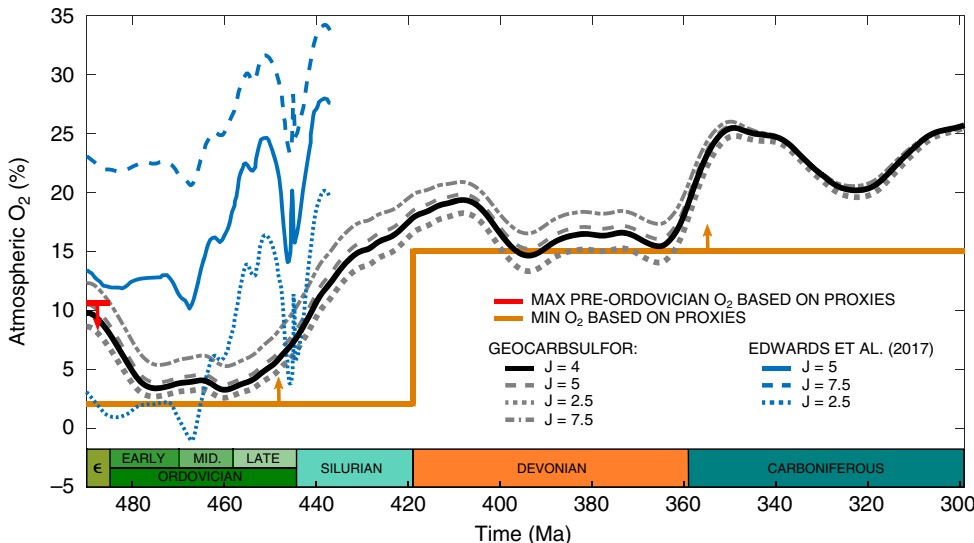

**Fig. 6** The effect of the parameter, J, on GEOCARBSULFOR oxygen outputs, compared to another method of calculating atmospheric oxygen. The black line is our baseline model run with J = 4. The grey dashed line is J = 5, the grey dotted line is J = 2.5, and the grey dot-dash line is J = 7.5 for our GEOCARBSULFOR model. The blue line is J = 5, the dashed line is J = 7.5 and the dotted line is J = 2.5 from Edwards et al.[55] The red line shows the approximate maximum atmospheric O$_2$ (10.4% atm) for the Cambrian and earlier, based on geochemical water column redox data and allowing for modelling uncertainties[14,72,73]. The orange line is the approximate O$_2$ minimum, based on the same redox data, as well as Cambrian biota and wildfire oxygen requirements[2,4,14]

δ$^{34}$S record, which may be compared to geological data. The model output does not reproduce the full degree of variability in measured δ$^{34}$S, but it does remain within, or very close to, the data range defined by the record, giving confidence that the modified sulphur cycle is an accurate representation (Supplementary Fig. 1).

Several uncertainties regarding O$_2$ levels are still inherent within the model; including the definitions of the mass fluxes and uncertainties in the carbon isotope record itself. The carbon cycle, primarily via organic carbon burial, in GEOCARBSULF is heavily dependent on the δ$^{13}$C record input into the model. The compilation by Saltzman and Thomas[48] provides a comprehensive set of data for all geologic time, but as they acknowledge, differences in the materials analysed (calcareous organisms, bulk versus component), sediment sources (epeiric seas vs. open oceans),

temperature, diagenetic alteration (in the case of high Mg calcite brachiopod shells) etc., can demonstrate variability in the isotopic signature they produce[32,62,63].

We conducted a sensitivity analysis to test the effect of variations in the δ$^{13}$C dataset on pO$_2$ predictions for our model (see Fig. 5). An initial attempt was made to run the model with a ±1 standard deviation to the δ$^{13}$C record, with all other parameters remaining as they were in our baseline run (GEOCARBSULFOR—red line in Fig. 5). The model deals adequately with the plus one standard deviation change, however, the minus one standard deviation results in model failure almost immediately, as pO$_2$ crashes to 0%. Although our treatment of the Saltzman and Thomas[48] data has smoothed out some of the more extreme δ$^{13}$C excursions, and our numerical method has made the model more resilient, the model may be missing some

additional stabilizing feedbacks that can counter particularly negative $\delta^{13}$C values, especially at model initiation. The model thus remains highly sensitive to the $\delta^{13}$C record used, but the general trend in oxygenation remains the same. Further revisions to the Saltzman and Thomas[48] compilation or a different treatment of the data may, however, result in significant differences to the computed evolution of Phanerozoic oxygen levels.

We performed a further sensitivity analysis, by investigating the effect of changes to the parameter, $J$, which is used in the model to alter the effects of oxygen concentration on carbon isotope fractionation (see supplementary equation (23)). In our baseline run, following the original model, we use $J = 4$. A recent study by Edwards et al.[55], has used this equation to estimate atmospheric oxygen concentrations in the early Paleozoic from the $\delta^{13}$C record in organic C and carbonates, using values for $J$ of 2.5 to 7.5. We compare our results in Fig. 6. GEOCARBSULFOR does not show as great a temporal variability in $pO_2$ levels during this period, and shows considerably less sensitivity to changes in the $J$ parameter, due to the much greater complexity of the model adding negative feedbacks that buffer against change. Nevertheless, both methods show an increase in atmospheric $O_2$ between the mid Ordovician and early Silurian. Additional sensitivity analyses and comparisons to other work were performed to confirm the robustness of low early Paleozoic $O_2$ and provide further validation of our model. These can be found in the Supplementary Note 1.

Overall, GEOCARBSULFOR shows a clear transition from an Earth with relatively low $pO_2$ (in terms of % atm), to a planet with an abundance of free atmospheric oxygen, in the process paving the way for the evolution of large terrestrial fauna. This well-defined, stepwise oxygenation—a Paleozoic Oxygenation Event (POE)—was coincident with the advent of terrestrial vascular plants, which fundamentally changed the geologic oxygen cycle. Our prediction of low $pO_2$ in the early Paleozoic removes a central disagreement between the results of GEOCARBSULF with those of other models, and between the model and geochemical proxies. More importantly, we develop here a robust, detailed model for Phanerozoic $O_2$ evolution that shows a number of potential links between atmospheric composition and biosphere evolution. This model provides a baseline against which to further investigate these ideas, and the opportunity to extend these approaches to investigate the Precambrian and the Neoproterozoic Oxygenation Event[16].

## Methods

**Model reconstruction.** The GEOCARBSULF[18,21] model was reconstructed in MATLAB and, using a four-stage implicit Runge-Kutta method, was solved numerically[64,65]. The Runge-Kutta method allows the model to generate outcomes with a high accuracy, whilst retaining stability, thus permitting updates to several parameters, with a reduction in model failure compared to the original model code[19,64,65].

**Data updates.** The $\delta^{13}$C record was updated to use data from Saltzman and Thomas[48], to which we applied a 10 Myr moving average, and the data was smoothed to eliminate variations in the short term. The $\delta^{34}$S record, which was required for testing our model, was updated to use data from Wu et al.[66]. The dimensionless parameters: land area ($f\_A$); river runoff ($f\_D$); proportion of land underlain by carbonates ($f\_L$); and effect of change of paleogeography on temperature ($GEOG$), as well as the temperature dependence parameter for carbonate, and silicate weathering ($gcm$), were revised to those employed by Royer et al.[19], some of which are based on the work of Goddéris et al.[67]. An additional parameter: the fraction of land experiencing chemical weathering ($f\_AW$), used by Royer et al.[19], was included, but we observe that their original data for this parameter was not normalized to the present, as all other parameters for GEOCARBSULF are, thus we have updated these values.

**Sulphur cycle.** The weathering and burial equations for young pyrite and gypsum were altered to those used in the COPSE model[24].

The weathering equations follow the reasoning that most gypsum is weathered with carbonate rocks, and most pyrite is weathered with silicate rocks, therefore

they can include a dependency on the carbonate and silicate weathering functions[68]. As COPSE doesn't utilize the rapid recycling inherent in GEOCARBSULF, changes were made to the young weathering fluxes. The ancient weathering fluxes remained as per GEOCARBSULF, but with an oxidative feedback function added to the weathering of ancient pyrite.

The pyrite burial equation was altered to eliminate the reliance on the sulphur isotope fractionation present in GEOCARBSULF, whilst retaining an oxygen dependent feedback on the amount of pyrite buried at each time step. As the model remains close to steady state throughout its run, the gypsum burial equation was modified to reduce the limitation the original equation imposed on pyrite burial. The new weathering and burial equations are as follows:

$$F\_wp\_y = \frac{F\_ws}{F\_ws\_0} \cdot new\_kwp \cdot \frac{Pyr\_y(t)}{Pyr\_0} \cdot O_2mr^{0.5} \qquad (3)$$

$$F\_wgyp\_y = \frac{F\_wc\_y}{F\_wc\_y0} \cdot new\_kwgyp \cdot \frac{Gyp\_y(t)}{Gyp\_0} \qquad (4)$$

$$F\_bp = k\_bp \cdot \frac{OA\_S(t)}{OA\_S\_0} \cdot \frac{F\_bg(t)}{F\_bg\_0} \cdot \frac{1}{O_2mr} \qquad (5)$$

$$F\_bgyp = k\_bgyp \cdot \frac{OA\_S(t)}{OA\_S\_0} \cdot Calc \qquad (6)$$

where:

$F\_wp\_y$ is the young pyrite weathering flux; $F\_wgyp\_y$ is the young gypsum weathering flux; $F\_bp$ is the pyrite burial flux; and $F\_bgyp$ is the gypsum burial flux.

$F\_ws$ is the silicate weathering flux at time ($t$); $F\_ws\_0$ is the present day silicate weathering flux; $new\_kwp$ is the pyrite weathering rate constant; $Pyr\_y(t)$ is the size of the young pyrite reservoir at time ($t$); $Pyr\_0$ is the size of the young pyrite reservoir at present day. $F\_wc\_y$ is the carbonate weathering flux at time ($t$); $F\_wc\_y0$ is the present day carbonate weathering flux; $new\_kwgyp$ is the gypsum weathering rate constant; $Gyp\_y(t)$ is the size of the young gypsum reservoir at time ($t$); and $Gyp\_0$ is the size of the young gypsum reservoir at present day. $OA\_S(t)$ is the size of the ocean sulphate reservoir at time ($t$); $OA\_S\_0$ is the size of the ocean sulphate reservoir at present day; $F\_bg(t)$ is the organic carbon burial flux at time ($t$); $F\_bg\_0$ is the organic carbon burial flux at present day; $k\_bp$ is the pyrite burial rate constant; and $k\_bgyp$ is the gypsum burial rate constant.

Calc is the normalized calcium reservoir[69], which is dimensionless, as is $O_2mr$: the normalized amount of $O_2$ in the ocean-atmosphere reservoir. All other reservoir sizes are in moles, and fluxes are in moles per Myr.

Finally, we make some alterations to the total amount of sulphur, and the apportioning of this to the various reservoirs, in the model. In the original GEOCARBSULF, the total amount of sulphur in the system is $638 \times 10^{18}$ moles. First, we reduce the total amount of sulphur in the system to $418 \times 10^{18}$ moles; we have conservation of mass in the model, so this reduction allows the model to run with a total sulphur value roughly equivalent to that used by Kump and Garrels[70]. Next, we challenge GEOCARBSULF's assumption that the initial sizes of the pyrite and gypsum reservoirs are equal to each other (combined young and ancient pyrite is $300 \times 10^{18}$ moles, as is the combined young and ancient gypsum). Following the work of Canfield and Farquhar[71], who provide evidence for a Proterozoic dominated by pyrite burial, with low gypsum deposition across the Ediacaran-Cambrian boundary, we adjust the balance of the sulphur distribution across the sedimentary reservoirs. We retain GEOCARBSULF's ocean-atmosphere reservoir value of $38 \times 10^{18}$ moles, and then start the model run at 570 Ma, with total gypsum equal to $100 \times 10^{18}$ moles, and total pyrite equal to $280 \times 10^{18}$ moles (see Supplementary Note 1 and Supplementary Fig. 6 for further information), thus changing the pyrite to gypsum ratio from 1:1 to 2.8:1.

**Changes to other fluxes and reservoirs.** The changes we made to the sulphur cycle resulted in the need to update other fluxes in the model. In the original GEOCARBSULF, degassing fluxes are contingent on spreading rates at time ($t$) multiplied by the present day rate, while ancient reservoirs are forced to remain at steady state throughout an entire model run. These formulations introduce a rigidity to the model's operations, which can be a source of failure, as the model cannot stabilize itself quickly enough following large perturbations. The following changes make the model more dynamic, allowing it to respond faster to fluctuations in the system.

We modified the original equations for the degassing of ancient reservoirs of pyrite, gypsum, organic carbon, and carbonate, so the degassing flux calculated at each time step was dependent on the total amount of material in each reservoir, multiplied by a rate constant and the spreading rate at time ($t$), with an additional dependence on the relative proportions of carbonates on shallow platforms or the deep ocean for carbonate degassing.

The weathering equations for ancient organic carbon and ancient carbonates were also updated: replacing the terms: $F\_wg\_a0$ and $F\_wc\_a0$—the modern day weathering fluxes for ancient organic carbon and ancient carbonates respectively—with a rate constant multiplied by the total amount of material in each reservoir at

each time step; we also include an oxidative feedback to the weathering equations for young and ancient organic carbon.

Finally, the equations governing the flux of material from young to ancient reservoirs at each iteration were altered, to allow the total amount stored in the ancient reservoirs to vary, instead of remaining constant over geologic time. This young to ancient flux is now dependent on the total amount stored in the respective young reservoir multiplied by a rate constant. The model remains in a steady state, but the total mass apportioned to each reservoir at each time step, by the model, has greater variance.

**Code availability**. The code for GEOCARBSULF, reconstructed in MATLAB, and the code for GEOCARBSULFOR (also MATLAB), both of which support this study, are available from the corresponding author upon request.

## Data availability

The datasets required to run the models, all of which support this study, are available from the corresponding author upon request.

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

## Acknowledgements

A.J.K. is funded by a studentship from the NERC SPHERES Doctoral Training Partnership (NE/L002574/1). B.J.W.M. is funded by a University of Leeds Academic Fellowship. S.Z. acknowledges financial support from Yale University. N.J.P acknowledges funding from Alternative Earths NASA Astrobiology Institute. S.W.P. and T.M.L. are supported by NERC (NE/P013651) and by Royal Society Wolfson Research Merit Awards.

## Author contributions

A.J.K. and B.J.W.M. designed the research. A.J.K. performed the modelling. A.J.K., B.J.W. M. and S.W.P. wrote the paper, with input from S.Z., N.J.P. and T.M.L.

## Additional information

**Competing interests:** The authors declare no competing interests.

