## [Peer Review File · Nature Communications]

Reviewers' comments:

Reviewer #1 (Remarks to the Author):

The manuscript 'stepwise oxygenation of the Paleozoic atmosphere' by Krause and co. addresses a topical problem in Earth evolution by modifying an existing model that outputs atmospheric oxygen through time. The results of the modelling exercise are interesting and of broad import, but I did not find the current form of the manuscript either compelling or engaging. I believe the work is well suited for Nature Comm. but could be expanded appreciably within the 5000 word scope of a Nature Comm. article. Specifically, it would be interesting and useful to present an expanded discussion of the difference between the two Geocarbsulf models as well as the COPSE model in the main text. It would also help to have results from sensitivity analyses in the main figures and discussed explicitly in the main text. Likewise, can more be done to tether the model results to the geological record? Can more of the nuances in the model results be discussed? It would be useful to plot the residuals between the two geocarbsulf models and explain specific differences. For example, there is an apparent strong decoupling at the Permo-Triassic boundary? Is this real? Can this be linked to a perturbation in the Earth System? Which model makes more sense? If these points can be addressed in an expanded version of the manuscript, I think this work will make an important contribution to our knowledge of the Phanerozoic Earth System and the history of atmospheric O₂.

Further comments:

Ln 54-55 Can you be more explicit on what this residence time is? For example, atmospheric O₂ residence time is only a few thousand years so where does the >1Myr number come from?

Generally, much of the new insight into S-isotope systematics is overlooked. While I agree that this represents uncertainty in the models, it is not true that this is an insurmountable problem and there are robust and effective approaches that address this. It would be good to expand this discussion and give some credit to these efforts.

A more detailed discussion of the connection between S-burial, weathering and oxygen would be useful.

Figures 1 and 3 are somewhat redundant, while Figure 2 is not particularly informative. I think better use of the figure space could be made.

Reviewer #2 (Remarks to the Author):

This manuscript presents some modifications to the classic Berner model (GEOCARBSULF) to better align it with other model estimates and the scant geological record that puts some constraint on O₂ levels. These changes, including a new way to treat the sulfur cycle and incorporating a new d¹³C record as a model driver, lead to oxygen levels that are more consistent with other model estimates and generate the lower O₂ levels for the Paleozoic that the authors prefer. The paper is well written and it's generally clear (especially after reading the supplement) what the authors have done to modify the model. In the end, I'm wondering how broadly interesting this result is, though. Certainly for those who model this time period, the reconciliation of these two somewhat different modeling approaches is comforting. In other words, I think this is a valuable contribution but I'm not sure it is properly suited for Nature Communications publication.

Some relatively minor comments and suggestions:

1) The authors point out that the model is run in quasi-steady state, in other words a series of

steady state algebraic expressions is solved. Given that, it's not obvious to me why they use an ODE solver (described in the Methods section). What ODEs are being solved; their should be none in a steady state model.

2) A little clearer explanation of why O₂ levels were higher or lower at certain times among different versions of their model would be helpful. Are O₂ levels lower in the Paleozoic because the authors start with a larger sedimentary pyrite reservoir? Are organic matter burial rates lower in the Pz as calculated from the Saltzmann and Thomas data? Etc.

Krause et al. Response to reviews

We thank the reviewers for their constructive comments, which have led to significant revision and improvement of the paper. Original comments are in black and our responses follow in blue.

Reviewer #1 (Remarks to the Author):

The manuscript 'stepwise oxygenation of the Paleozoic atmosphere' by Krause and co. addresses a topical problem in Earth evolution by modifying an existing model that outputs atmospheric oxygen through time. The results of the modelling exercise are interesting and of broad import, but I did not find the current form of the manuscript either compelling or engaging. I believe the work is well suited for Nature Comm. but could be expanded appreciably within the 5000 word scope of a Nature Comm. article.

- Specifically, it would interesting and useful to present an expanded discussion of the difference between the two Geocarbsulf models as well as the COPSE model in the main text.

We have expanded our discussion of the difference between the original GEOCARBSULF and our revised version, and between GEOCARBSULF modelling and the COPSE model (lines: 56 – 69; 285 – 312; 315 – 325). We have also added Figure 4 which further highlights the differences between the original GEOCARBSULF and our revised model.

- It would also help to have results from sensitivity analyses in the main figures and discussed explicitly in the main text.

We have moved sensitivity analysis figures and their discussion from the SI to the main text and added discussion (these are now figures 5 and 6, plus lines: 340 – 380).

- Likewise, can more be done to tether the model results to the geological record? Can more of the nuances in the model results be discussed?

We have added more detail on the geological record. Figure 1 now compiles geochemical evidence for Paleozoic oxygenation, and the revised Figure 3 compares the model O₂ predictions to paleobiological and proxy evidence in an expansion of the results and discussion section. The expansion of the results and discussion section (lines: 216 – 283), the revised figure 3, and the new figures 4 to 6 describe some of the nuances in the model results.

- It would be useful to plot the residuals between the two geocarbsulf models and explain specific differences. For example, there is an apparent strong decoupling at the Permo-Triassic boundary? Is this real? Can this be linked to a perturbation in the Earth System? Which model makes more sense?

We tried plotting the residuals as suggested but found that a simple comparison of the new and old models (Fig 3) was a more informative illustration on which to base our discussion of the differences between models. We have expanded our discussion of the differences between the original GEOCARBSULF O₂ reconstruction and our modified version, aided by the new Figure 4 (and lines: 285 – 325), which demonstrates which of these differences come from using the updated δ¹³C record versus the updated model sulphur cycle. For the Permo-Triassic, the original GEOCARBSULF assumed a very steep drop in global average δ¹³C, with low values continuing for much of the Triassic and Jurassic. These patterns are not present in recent δ¹³C compilations, which show a much more stable value through this timeframe, hence the differences in models at this time.

If these points can be addressed in an expanded version of the manuscript, I think this work will make an important contribution to our knowledge of the Phanerozoic Earth System and the history of atmospheric O₂.

Further comments:

- Ln 54-55 Can you be more explicit on what this residence time is? For example, atmospheric O₂ residence time is only a few thousand years so where does the >1Myr number come from?

We have amended our paper to state more clearly that we mean the residence time of O₂ in the combined atmosphere and ocean system vs the sediments/lithosphere.

Lines 53-55 said, 'GEOCARBSULF, makes predictions by estimating source and sink fluxes in the geological oxygen cycle, which is possible because the residence time of O₂ in the surface system is very long (>1 Myr).'

We have changed the line (52 – 53) to:

'...O₂ in the surface system (i.e. atmosphere and ocean) is very long (>1 Myr).'

e.g. the atmosphere-ocean oxygen reservoir is 3.8×10^{19} mol and the source flux is 1.8×10^{13} mol, dividing reservoir by source gives a residence time of ~2.1 Myrs.¹

- Generally, much of the new insight into S-isotope systematics is overlooked. While I agree that this represents uncertainty in the models, it is not true that this is an insurmountable problem and there are robust and effective approaches that address this. It would be good to expand this discussion and give some credit to these efforts.

We have included some more information in this section (lines: 172 – 180) and clarified our view that there remains a great deal of uncertainty specifically with regards to the relationship between sulphur isotope fractionation and atmospheric oxygen levels, particularly on a global scale and through geologic time. We are not aware of any further specific details that we have not covered that are of relevance to this kind of modelling.

- A more detailed discussion of the connection between S-burial, weathering and oxygen would be useful.

We have added equations and additional lines of text (lines 123-127) to expand this section.

- Figures 1 and 3 are somewhat redundant, while Figure 2 is not particularly informative. I think better use of the figure space could be made.

We have considerably revised figures 1, 2 and 3, so that figure 1 focuses on GEOCARBSULF's results in the Paleozoic and compares it to a suite of geochemical proxy data; figure 2 is more informative, as it now highlights explicitly the feedback loop in the model caused by the formulation for sulphur isotope fractionation; and figure now 3 compares our results to the original GEOCARBSULF and COPSE models, as well as paleobiological data and important geobiological events in the Paleozoic. In line with above comments, we have also expanded the manuscript to include 6 figures in the main text.

Reviewer #2 (Remarks to the Author):

This manuscript presents some modifications to the classic Berner model (GEOCARBSULF) to better align it with other model estimates and the scant geological record that puts some constraint on O₂ levels. These changes, including a new way to treat the sulfur cycle and incorporating a new d¹³C record as a model driver, lead to oxygen levels that are more consistent with other model estimates and generate the lower O₂ levels for the Paleozoic that the authors prefer. The paper is well written and it's generally clear (especially after reading the supplement) what the authors have done to modify the model. In the end, I'm wondering how broadly interesting this result is, though. Certainly for those who model this time period, the reconciliation of these two somewhat different modeling approaches is comforting. In other words, I think this is a valuable contribution but I'm not sure it is properly suited for Nature Communications publication.

We have modified a significant amount of text in our paper in line with this review to make much more clear the importance and broad appeal of the work. To summarize here, we note that:

- Making accurate atmospheric oxygen estimates for the Paleozoic is a major challenge for the geoscience community, as knowledge of surface oxygen levels at this time are required to address the fundamental question of the influence of oxygen on animal evolution (e.g. Lu et al.²)
- GEOCARBSULF currently provides the most highly-detailed, proxy-driven estimate of Phanerozoic atmospheric O₂ variation, and therefore the GEOCARBSULF reconstruction (or a simplification of it) is commonly used to assess relationships between oxygen and animal evolution in very high-profile work (e.g. [3–6])
- We have described and fixed a major structural problem in the GEOCARBSULF model that gave demonstrably unrealistic results, leading to a complete revision of the model O₂ estimates that shows a considerable difference to the original model.
- This revision indeed fixes the conflict with the sparse Paleozoic proxy record and other types of models, but the more important result is our generation of a new detailed prediction of atmospheric oxygen evolution for the community, and a further independent line of support for the hypothesis that atmospheric O₂ rose during the Paleozoic, influencing animal evolution.

Some relatively minor comments and suggestions:

- The authors point out that the model is run in quasi-steady state, in other words a series of steady state algebraic expressions is solved. Given that, it's not obvious to me why they use an ODE solver (described in the Methods section). What ODEs are being solved; their should be none in a steady state model.

We are not solving a system of steady state algebraic equations here (which would require rewriting the model such that $dy/dt = 0$ for each reservoir and solving from there). The original GEOCARBSULF⁷⁻¹¹ as well as GEOCARB¹²⁻¹⁴ and other early oxygen modelling^{15,16} are systems of ODEs that tend towards steady states when run forwards in time. We have used the same approach in our work. We have cleared up the text that may have misled here and thank the reviewer for pointing this out.

- A little clearer explanation of why O₂ levels were higher or lower at certain times among different versions of their model would be helpful. Are O₂ levels lower in the Paleozoic because the authors start with a larger sedimentary pyrite reservoir? Are organic matter burial rates lower in the Pz as calculated from the Saltzman and Thomas data? Etc.

We have revised previous Figure A2 to become Figure 4 to help answer the question about the roles of the revised sulphur cycle and revised carbon isotopes in producing lower oxygen levels in the Paleozoic. Paleozoic O₂ is lower because the new model sulphur cycle buries significantly less pyrite than the old model at this time. Both models bury less carbon in the Paleozoic than the present day, but low Paleozoic O₂ is robust to the C isotope record used (see figure 4). The size of the sedimentary pyrite reservoir in the Paleozoic is not as important a factor here, and is set by the requirement that the model reproduces present day reservoir size at the end of the run.

References

1. Catling, D. C. & Zahnle, K. J. in *Encyclopedia of Atmospheric Sciences* (eds. Holton, J. R., Curry, J. A. & Pyle, J. A.) 754–761 (Academic Press, 2002).
2. Lu, W. *et al.* Late inception of a resiliently oxygenated upper ocean. **5372**, 1–8 (2018).
3. Falkowski, P. G. *et al.* The Rise of Oxygen over the Past 205 Million Years and the Evolution of Large Placental Mammals. *Science* **309**, 2202–2204 (2005).
4. Lyons, T. W., Reinhard, C. T. & Planavsky, N. J. The rise of oxygen in Earth's early ocean and atmosphere. *Nature* **506**, 307–15 (2014).
5. Clapham, M. E. & Karr, J. A. Environmental and biotic controls on the evolutionary history of insect body size. *Proc. Natl. Acad. Sci.* **109**, 10927–10930 (2012).
6. Edwards, C. T., Saltzman, M. R., Royer, D. L. & Fike, D. A. Oxygenation as a driver of the Great Ordovician Biodiversification Event. *Nat. Geosci.* **10**, 925–930 (2017).
7. Berner, R. A. GEOCARBSULF: A combined model for Phanerozoic atmospheric O₂ and CO₂. *Geochim. Cosmochim. Acta* **70**, 5653–5664 (2006).
8. Berner, R. A. Inclusion of the weathering of volcanic rocks in the GEOCARBSULF model. *Am. J. Sci.* **306**, 295–302 (2006).

9. Berner, R. a. Addendum to 'Inclusion of the Weathering of Volcanic Rocks in the GEOCARBSULF Model': (R. A. Berner, 2006, V. 306, p. 295-302). *Am. J. Sci.* **308**, 100–103 (2008).
10. Berner, R. A. Phanerozoic atmospheric oxygen: new results using the geocarbsulf model. *Am. J. Sci.* **309**, 603–606 (2009).
11. Royer, D. L., Donnadieu, Y., Park, J., Kowalczyk, J. & Godd ris, Y. Error analysis of CO₂ and O₂ estimates from the long-term geochemical model GEOCARBSULF. *Am. J. Sci.* **314**, 1259–1283 (2014).
12. Berner, R. A. A model for Atmospheric CO₂ over Phanerozoic Time. *Am. J. Sci.* **291**, 339–376 (1991).
13. Berner, R. A. 3GEOCARBII: A Revised Model of Atmospheric CO₂ over Phanerozoic Time. *Am. J. Sci.* **294**, 56–91 (1994).
14. Berner, R. a. and Kothavala, Z. GEOCARB III; a revised model of atmospheric CO₂ over Phanerozoic time. *Am. J. Sci.* **301**, 182–204 (2001).
15. Berner, R. A. Models for Carbon and Sulfur Cycles and Atmospheric Oxygen: Application to Paleozoic Geologic History. *Am. J. Sci.* **287**, 177–196 (1987).
16. Berner, R. A. Modeling atmospheric O₂ over Phanerozoic time. *Geochim. Cosmochim. Acta* **65**, 685–694 (2001).

REVIEWERS' COMMENTS:

Reviewer #2 (Remarks to the Author):

I feel that the authors have done a nice job responding to my comments and those of the other reviewer. I recommend publication of the manuscript.